# Investigation of Alogliptin-Loaded In Situ Gel Implants by 2^3^ Factorial Design with Glycemic Assessment in Rats

**DOI:** 10.3390/pharmaceutics14091867

**Published:** 2022-09-05

**Authors:** Tarek M. Ibrahim, Margrit M. Ayoub, Hany M. El-Bassossy, Hanan M. El-Nahas, Eman Gomaa

**Affiliations:** 1Department of Pharmaceutics, Faculty of Pharmacy, Zagazig University, Zagazig 44519, Egypt; 2Department of Pharmacology and Toxicology, Faculty of Pharmacy, Zagazig University, Zagazig 44519, Egypt

**Keywords:** alogliptin, PLGA, in situ gel implants, factorial design, blood glucose level

## Abstract

The aim of the study was to design injectable long-acting poly (lactide-co-glycolide) (PLGA)-based in situ gel implants (ISGI) loaded with the anti-diabetic alogliptin. Providing sustained therapeutic exposures and improving the pharmacological responses of alogliptin were targeted for achieving reduced dosing frequency and enhanced treatment outputs. In the preliminary study, physicochemical characteristics of different solvents utilized in ISGI preparation were studied to select a proper solvent possessing satisfactory solubilization capacity, viscosity, water miscibility, and affinity to PLGA. Further, an optimization technique using a 2^3^ factorial design was followed. The blood glucose levels of diabetic rats after a single injection with the optimized formulation were compared with those who received daily oral alogliptin. N-methyl-2-pyrrolidone (NMP) and dimethyl sulfoxide (DMSO), as highly water-miscible and low viscous solvents, demonstrated their effectiveness in successful ISGI preparation and controlling the burst alogliptin release. The impact of increasing lactide concentration and PLGA amount on reducing the burst and cumulative alogliptin release was represented. The optimized formulation comprising 312.5 mg of PLGA (65:35) and DMSO manifested a remarkable decrease in the rats’ blood glucose levels throughout the study period in comparison to that of oral alogliptin solution. Meanwhile, long-acting alogliptin-loaded ISGI systems demonstrated their feasibility for treating type 2 diabetes with frequent dosage reduction and patient compliance enhancement.

## 1. Introduction

As declared by World Health Organization (WHO), diabetes disease was considered the ninth direct cause of death among people worldwide in 2019 [1]. In particular, type 2 diabetes may be diagnosed many years later after its onset and the appearance of its complications [2]. Therefore, the importance of patient-centered strategies for the management of hyperglycemia in type 2 diabetes is continuously reinforced by governmental authorities and the research community [3]. Tailoring glycemic targets and glucose therapies to patients besides the role of diet and exercise are followed as a proper approach to help improve patient compliance and adherence to treatment [4].

Dipeptidyl peptidase (DPP)-4 inhibitors are a class of drugs prescribed to control hyperglycemia in type 2 diabetes patients. The DPP-4 enzyme rapidly deactivates the incretin hormones released after meals, such as glucose-dependent insulinotropic polypeptide and glucagon-like peptide-1. The incretin levels can then be increased, by antagonizing the DPP-4, resulting in higher insulin synthesis and release, lower glucagon secretion, and in turn regulated glucose homeostasis [5,6]. With specific attention to alogliptin as a DPP-4 inhibitor, the U.S. Food and Drug Administration (FDA) approved three new oral alogliptin-based agents in hyperglycemia treatment, including alogliptin alone [7], alogliptin-metformin hydrochloride combination [8], and alogliptin-pioglitazone combination [9]. Oral alogliptin tablets are available in the form of three doses (6.25, 12.5, and 25 mg). If the patient has moderate or severe renal impairment, the dose can be decreased [10].

Although the oral route is the most convenient method for the delivery of drugs, it has several limitations, including the likelihood of forgetting dosage, early discontinuation, fluctuations in plasma drug levels caused by daily medication regimens, frequent administration of the drug, enzymatic degradation, and low drug absorption resulting in the occurrence of side effects and poor patient compliance [11,12]. These issues can be mitigated by developing modified release dosage forms of drugs under consideration [13]. Subsequently, long-acting injectable formulations gain much interest as they can provide sustained therapeutic exposure to drugs, reduced dosing frequency, and enhanced treatment outputs [14]. These preparations present various merits over traditional formulations in terms of providing weekly, monthly, or even annually sustained release of drugs in addition to the achievement of therapeutic drug action at lower concentrations. This can help reduce the dosage frequency, diminish the undesirable adverse effects, improve the pharmacological responses and fulfill patient adherence and therapeutic outcomes [15,16].

For instance, micro and nanoparticle systems are one of the most widespread promising delivery systems that can be injected into the body using conventional needles. These carriers are extensively investigated owing to the fact that the desired release pattern of the drug and its biodistribution can be improved by modulating their surface properties [17]. They can also protect the drug from degradation, enhance its absorption, and withstand physiological stress and biological stability [18]. In comparison to the required deep intramuscular injection, syringe clogging or possible stability troubles under certain conditions of such carriers, the long-acting injectable in situ gel implant (ISGI) systems are recognized as ideal substitutes for sustaining the release of drugs [15,19]. The ISGI systems are characterized by their minimally invasive administration, compatibility with several drugs, and simple inexpensive processing, including only dissolution or suspension of the drug in a polymeric solution [20,21]. Moreover, they are promising approaches for local and systemic delivery of single and multiple drugs [22]. Hence, the ISGI systems are favorably used to produce more suitable drug-loaded candidates for the treatment of different diseases, such as prostate cancer [23], schizophrenia [24], asthma [25], inflammation [26], hyperlipidemia [27], opioid use [28], etc.

The mechanism of action of ISGI systems depends on the solution–gel conversion technique. The formulation is firstly injected as a low viscous solution and then converted into a rigid implant or depot upon being injected into the body [29]. The ISGI systems are triggered through different transformation mechanisms, such as in situ cross-linking [30], in situ solidifying organogels [31], or in situ phase separation. The phase separation may be based on various factors, for example, temperature change [32], pH change [33], solvent exchange [34] or light [35]. Solvent exchange-induced phase separation mechanism is commonly used in drug delivery research as it can be easily triggered requiring only an aqueous environment [36]. The biodegradable polymer is dissolved in a biocompatible organic solvent. Upon injecting the polymeric solution containing the drug into the body, the organic solvent diffuses into the surrounding aqueous phase and water or tissue fluid (non-solvent) penetrates the ISGI causing phase separation and in situ precipitation of the hydrophobic polymer. Eventually, the solid ISGI system is established at the site of injection having the ability to sustain the release of the drug [37,38].

As one of the most widespread phase-sensitive smart polymers, the poly (lactide-co-glycolide) (PLGA) polymer occupies a rapidly growing area in the biomedical and pharmaceutical fields. It has been approved by the U.S. FDA and European Medicine Agency (EMA) by the virtue of its premium biocompatibility, biodegradability, safety, and sustained release characteristics [39]. In addition, PLGA is safely degraded in the body as its monomeric end products are readily eliminated through the Krebs cycle [40]. The polymer is commercially available in form of various grades with different lactide/glycolide ratios, molecular weights, inherent viscosities, and end cappings. Therefore, the choice of a suitable grade of PLGA polymer along with convenient solvent and concentration of each component added with the studied drug is certainly important for the optimization of drug release [15]. Because the PLGA polymer is not soluble in water, it is solubilized in organic solvents for the preparation of ISGI systems. Using the solvent exchange-induced phase separation technique, the used solvent must have low viscosity, high water affinity, the ability to dissolve PLGA polymer and non-toxicity [41]. For examples of widely used solvents, N-methyl-2-pyrrolidone (NMP), dimethyl sulfoxide (DMSO), propylene glycol (PG), benzyl alcohol (BA) and 2-pyrrolidone are commonly utilized as water-miscible solvents, while ethyl acetate (EA), triacetin (TA) and benzyl benzoate (BB) are commonly utilized as non-miscible solvents [42,43]. Despite the various merits of the ISGI systems, the initial burst release of the loaded drug during the implant development is still a great challenge that the ISGI systems encounter. It is an undesirable phenomenon where the drug is highly released within a few minutes or hours after ISGI administration and can occur due to several reasons [44]. Additionally, local or systemic adverse effects may exist owing to the high drug concentration in the blood following the ISGI injection [29].

The selection and optimization of various formulations are traditionally performed following the most used method called the one factor at a time (OFAT) experiment [45]. The OFAT depends on changing one factor at a time and it is expensive and time-consuming. The experiment is not properly designed and the data analysis cannot recompense the absence of planning. Thus, this approach may possibly lead to false conditions and inaccurate results owing to ignoring the impact of interactive factors [46,47]. From this perspective, the design of experiments (DoE) strategy is an organized system for determining the relationship between factors that affect the outputs and looking for better quality risk management [48]. It commences with predefined objectives and deciding target product profile with preferable manufacturing, efficacy and safety maintenance. Therefore, it is necessary to study the effect of different formulation factors on the quality of the preparation by using a minimal number of experimental runs followed by a subsequent selection of factors to develop an optimized formulation using established statistical tools for optimization [49,50]. Among the diverse experimental designs, the time-saving two-level factorial design is most commonly pursued where all the factors are studied in all possible combinations [51,52]. This design is regarded as the most efficient in evaluating the influence of individual factors and their interactions on the studied responses in addition to identifying what level of these factors can present better and desired responses [53].

The aim of our study was to study the impact of using several organic solvents possessing variable physicochemical characteristics on the successful preparation of ISGI systems loaded with anti-diabetic alogliptin and the appropriate control on the initial burst and cumulative release of the drug. The 2^3^ factorial design was followed in order to optimize an ISGI formulation of a sustained release capable of providing suitable therapeutic exposures of alogliptin and improving its hypoglycemic responses along with achieving reduced dosing frequency and enhanced treatment outputs.

## 2. Materials and Methods

### 2.1. Materials

Alogliptin was kindly supplied from Hikma Pharmaceuticals, 6 October, Egypt. PLGA polymers (65:35 and 85:15) were purchased from LACTEL Absorbable Polymers, Birmingham, AL, USA. NMP was purchased from Central Drug House, New Delhi, India. Triacetin was supplied from Euromedex, Souffelweyersheim, France. DMSO and ethyl acetate were purchased from SD Fine Chem Limited, Mumbai, India. PG and fructose were purchased from El-Nasr Pharmaceutical Chemicals, Cairo, Egypt. BA was purchased from Research-Lab Fine Chem Industries, Mumbai, India. Polyethylene glycol 400 (PEG 400) was purchased from Fluka Chemie AG, Buchs, Switzerland. BB was purchased from Alpha Chemika, Mumbia, India. Streptozotocin was purchased from Sigma Aldrich, St. Louis, MO, USA. All other chemicals were of analytical grade.

### 2.2. Methods

#### 2.2.1. Preliminary Study for Selecting Solvents Used in Preparation of ISGI Formulations

##### Saturation Solubility Measurement of Alogliptin in Different Organic Solvents

Measurement of saturation solubility of alogliptin was performed in various organic solvents. The solvents selected for the preliminary study were NMP, BA, DMSO, PEG 400, PG, EA, TA, and BB. In this test, excess amounts of pure alogliptin were suspended in each solvent (3 mL) in a separate glass vial. The vials were then transferred to a water bath shaker and kept shaken at 25 ± 0.5 °C and 100 rpm. Until reaching the equilibrium after 3 days, the contents of the vials were separately filtered using 0.22 μm nylon syringe filters and the filtrates were suitably diluted by methanol. Then, they were analyzed for measuring the amount of dissolved drug using ultraviolet-visible (UV-VIS) spectrophotometer at λ_max_ 278 nm after appropriate dilutions using the corresponding medium as a blank. The experiment was carried out in triplicate and the results were presented as mean values ± standard deviation (SD).

##### Preparation of Alogliptin-Loaded ISGI Formulations

In stoppered glass vials, the ISGI formulations containing alogliptin were simply prepared by dissolving the PLGA (85% lactide:15% glycolide) polymer in different organic solvents at a constant solvent to polymer ratio of 2:1, respectively. The vials were placed on a heating magnetic stirrer and the contents were mixed at 900 rpm and 60 °C for 30 min. Thereafter, alogliptin (62.5 mg) was added to the polymeric solution with a persistent mixing until the gel formulations were formed. The amounts of PLGA, solvent, and drug were set to generate 1 g total formulation. The prepared ISGI formulations were left to cool to room temperature and then kept in a refrigerator overnight for further studies.

##### In Vitro Release Study of Alogliptin-Loaded ISGI Formulations

The cumulative release of alogliptin from the prepared ISGI formulations was studied using a water bath shaker and receptor medium of 15 mL of phosphate buffer (pH 7.4) to fulfill the required sink conditions. The receptor medium was added to the prepared ISGI formulations in stoppered glass vials. The vials were kept shaken in the water bath shaker at 37 ± 0.5 °C and 100 rpm. The samples (0.5 mL) were taken at different time intervals on the first day to study the burst release of alogliptin and then were withdrawn every day afterward to study the cumulative drug release. An equal volume of fresh phosphate buffer was turned back to the vials to keep the volume of receptor medium constant. The samples were separately filtered by 0.22 μm nylon syringe filters and analyzed for measuring the amount of the released drug using UV-VIS spectrophotometer at λ_max_ 278 nm after appropriate dilutions using the corresponding medium as a blank. The mean values of the cumulative release of alogliptin-loaded ISGI formulations were compared to those of pure alogliptin solution. The experiment was performed three times and the results were presented as mean values ± SD.

#### 2.2.2. Experimental Design and Statistical Analysis

Experimental design is recognized as a systematic and scientific technique for studying the relation and interaction between the independent variables (factors) and the dependent variables (responses) [50]. In our study, Design-Expert^®^ software (version 11, Stat-Ease Inc, Minneapolis, MN, USA) was used for the experimental design, mathematical treatments, and statistical data analysis. A 3-factor, 2-level (2^3^) full factorial design was constructed with two replications of experimental runs to minimize the error in the design [54]. The design role was to optimize the selected formulation factors and estimate the main effects and the interaction effects on the studied responses. After the preliminary study, the selected factors were lactide concentration in PLGA (A), type of solvent (B), and PLGA amount (C). The studied responses were burst release of alogliptin after 6 h (Y_1_) and cumulative release of alogliptin after 10 days (Y_2_). Table 1 shows the two different levels of the tested factors and the goal required for the studied responses. Thereafter, sixteen runs were suggested and randomly ordered by the design software as shown in Table 2. The suggested formulations were prepared following the same abovementioned procedures (Preparation of Alogliptin-Loaded ISGI Formulations) and their in vitro initial and cumulative release % values were measured as previously described in the preliminary study (In Vitro Release Study of Alogliptin-Loaded ISGI Formulations). After entering the results of the responses, the following polynomial equation was used to fit the data values:Y = b_0_ + b_1_A + b_2_B + b_3_C + b_4_AB + b_5_AC + b_6_BC + b_7_ABC
where, Y is the measured response, (b_0_) is the intercept of the polynomial equation, (b_1_–b_7_) are the regression coefficients computed from the observed experimental values of the measured response, (A, B and C) are the main effects, (AB, AC and BC) are the two-way interaction effects, and (ABC) is the three-way interaction effects.

This polynomial equation would obtain the measured response and determine which factor and/or interaction could affect the response most [55]. The main effects could elucidate how changing the value of one factor from low to high could influence the measured response. Besides, the interaction effects could represent the average effect of changing one or more factor values on the measured response [51]. The polynomial equation could draw conclusions by considering the magnitude of coefficients where the positive sign reflected a synergistic effect and the negative sign meant an antagonistic effect [56].

Half-normal plots, Pareto charts, and the % contribution of factors to the responses were illustrated. The analysis of variance (ANOVA) and *p*-values with a 95% confidence interval (*p* < 0.05) were followed to decide the significance of each coefficient term. To evaluate the best fitting extent of data, various statistical parameters were compared, such as coefficient of variation (C.V.%), multiple correlation coefficient (R^2^), adjusted R^2^, predicted R^2^, adequate precision, and predicted residual sum of the square (PRESS). The normal plots of residuals and the graphs of residual versus experimental runs were constructed to appreciate the model’s adequacy. The perturbation, one factor, and interaction plots were obtained to study the relationships and interactions between the studied factors and the measured responses.

#### 2.2.3. Optimization Process and Statistical Validation

After the statistical analysis, a numerical optimization technique was followed for optimizing the formulation factors for the preparation of ISGI systems. The desirability function, as a commonly used multi-criteria methodology, could obtain the best-fitted levels of the factors and acquire the favorable response that could correspond to the goal criteria [57]. Based on the goal criteria required for the dependent responses as shown in Table 1, the design software could propose an optimized formulation with a desirability value from zero to one. When the desirability value tended towards one, this pointed out the compliance of the tested response to its ideal value [58]. Subsequently, the optimized formulation was evaluated with a quantitative comparison of the resulting experimental values of the responses with the values predicted by the design to calculate the % prediction error according to the following equation [59]:$$\%\;\mathrm{prediction}\;\mathrm{error}=\frac{\mathrm{predicted}\;\mathrm{value}\;-\;\mathrm{experimental}\;\mathrm{value}\;}{\mathrm{predicted}\;\mathrm{value}}\times 100$$

In order to investigate the mechanism of release kinetics of the optimized ISGI formulation, the data were subjected to the following models:Zero order model: Q_t_ = K_o_.t,First order model: Q_t_ = 1 − e^−Kt^,Higuchi model: Q_t_ = K_H_.t^1/2^,Hixson–Crowell model: Q_o_^1/3^ −/Q_t_^1/3^ = K_HC_.t, andKorsmeyer–Peppas model: Q_t_/Q_∞_ = K_KP_.t^n^where Q_t_ is the amount of drug released at time (t), Q_o_ is the initial amount of drug released, Q_∞_ is the amount of drug released at time infinity (∞), K_o_, K, K_H_, K_HC_, K_KP_ are the release rate constants of the previous models, respectively, and n is the release exponent [60]. The model of highest correlation coefficient (R^2^) was able to best describe the release mechanism of alogliptin from the optimized ISGI formulation. Then, the release mechanism was confirmed by the n value of Korsmeyer–Peppas model.

#### 2.2.4. In Vitro Characterization of Optimized ISGI Formulation

##### Scanning Electron Microscopy (SEM)

The internal structure and the external surface of the optimized alogliptin-loaded ISGI formulation were inspected by using a scanning electron microscope. The preparation was injected with a phosphate buffer (pH 7.4) in stoppered glass vials. It was kept shaken in the water bath shaker at 37 ± 0.5 °C and 100 rpm. Different samples of the optimized formulation were collected at different time intervals (0, 1, and 10 days) and allowed to dry at room temperature. Samples were spread as thin layers on aluminum stubs and then coated with gold using a sputter coater under a high vacuum. The SEM photomicrographs were taken at 250× magnification power at an accelerating voltage of 20–30 kV.

##### Fourier Transform Infrared (FTIR) Spectroscopy

The FTIR spectrums of pure alogliptin powder, PLGA polymer, physical mixture, selected solvent, alogliptin-free ISGI formulation, and optimized alogliptin-loaded ISGI formulation were scanned by an FTIR spectrophotometer following the potassium bromide disk method. The scanning range was 4000–500 cm^−1^ and the resolution was 1 cm^−1^.

##### Differential Scanning Calorimetry (DSC)

The thermal behaviors of pure alogliptin powder, PLGA polymer, physical mixture, selected solvent, alogliptin-free ISFI formulation, and optimized alogliptin-loaded ISGI formulation were studied using a DSC instrument. The samples were heated in sealed aluminum pans within the temperature range of 0–220 °C at a constant heating rate of 10 °C/min and under a nitrogen atmosphere with a flow rate of 30 mL/min.

#### 2.2.5. In Vivo Characterization of Optimized ISGI Formulation

##### Animals and Ethical Approval

Adult albino male rats weighing 200~250 g each were utilized. The animals were obtained from the animal breeding center, Zagazig University, Egypt. They were kept for one week before starting the experiments in a 12 h light and 12 h dark cycle at room temperature with free water and food. The experiments were conducted corresponding to the Institutional Animal Care and Use Committee (IACUC) guidelines of the Faculty of Pharmacy, Zagazig University (Approval number: ZU-IACUC/3/F/157/2021).

##### Induction of Diabetes in Rats

For induction of diabetes in rats, they received fructose (10%) in drinking water, NaCl (3%), and a high-fat diet (25%) in the food pellets for three weeks and then received a single intraperitoneal injection of streptozotocin (40 mg/kg). One week after streptozotocin injection, rats with stable postprandial hyperglycemia (200–300 mg/dL) were considered diabetic.

##### Animal Groups

Rats were randomly divided into four groups (n = 10) as follows:**Group 1:** Control rats received regular tap water and food for six successive weeks.**Group 2:** Hyperglycemia-induced rats received the vehicle for another two weeks.**Group 3:** Hyperglycemia-induced rats received daily oral alogliptin solution (2.5 mg/kg) for 10 days.**Group 4:** Hyperglycemia-induced rats received a single subcutaneous injection of the optimized alogliptin-loaded ISGI preparation (25 mg/kg).

##### Measurement of Postprandial Blood Glucose in Rats

The blood glucose levels were monitored at 1, 3, 7, and 10 days following the oral glucose load in all experimental groups. The rats were fasted for 8 h and then were given a constant glucose load (0.5 g). After one hour, the blood glucose levels were measured by a glucose meter from the tail droplet. The experimental parameters were analyzed by two-way ANOVA followed by Tukey’s post hoc test using GraphPad Prism^®^ program, version 5.00.

## 3. Results and Discussion

### 3.1. Preliminary Study for Selecting Solvents Used in Preparation of ISGI Formulations

For preparation of ISGI systems, selection of organic solvents, as main components in ISGI preparations, having convenient characteristics is crucial, especially in terms of safety, biocompatibility, viscosity, ability to dissolve polymers, and avoidance of local irritation or adverse effects at the injection site [41]. In our study, the utilized solvents are reported to be safe when used as additives in human parenteral products in addition to their high median lethal doses [25,61]. The solubility measurements of alogliptin in different organic solvents were carried out using water-miscible solvents (NMP, DMSO, BA, PG and PEG 400) and non-miscible solvents (EA, TA and BB) as presented in Table 3. The results showed that water-miscible solvents had a better solubilization capacity of alogliptin than that of non-miscible solvents. This could be attributed to the high solubility of the drug in water [62,63].

The chosen solvent should be capable of solubilizing the used biodegradable polymer. Upon preparing the ISGI formulations under the preliminary study, the PLGA polymer (85:15) was attempted for being dissolved in various organic solvents as shown in Table 3, where NMP, BA, DMSO and EA displayed good solubilization capacities toward PLGA, while other solvents (PEG 400, PG, TA and BB) demonstrated low and slow solubilization performance with incomplete PLGA solubilization. These findings might be ascribed to the former solvents being known as good solvents owing to their high affinity to the polymer and their small molecular volume. Camargo et al. [65] reported that solvents with smaller molecular volumes could represent higher penetration into the polymeric chain, hence enhancing the dissolution and prospective solidification of the PLGA polymer. Therefore, such solvents were selected for further investigation owing to their favorable characteristics along with their desirable low viscosity values (Table 3). However, although PG had a smaller molecular volume, it was excluded from further studies because it showed incomplete PLGA solubilization in the trials, possibly due to its higher viscosity value (58.10 cP). This could be consistent with the results reported by Kim et al. [66] who notified the incomplete solubility of PLGA polymer in PG solvent. Besides, PEG 400, TA and BB were excluded not only due to their higher molecular volume and incomplete solubilization of PLGA polymer but also owing to their high viscosity values. Garner et al. [67] reported that PEG 400 showed higher solubility values only for PLGA with lower lactide concentration and this was consistent with our results where PEG 400 did not demonstrate any solubilization performance on the studied PLGA containing 85% lactide. On another side, the increased viscosity of the polymer-solvent preparation was markedly seen during the dissolution of PLGA in TA or BB solvents followed by the imperfect solubilization of PLGA. Voigt [68] pointed out that low viscous solvents, such as NMP, BA, and EA could dissolve the polymer within 1–50 min at different PLGA concentrations, while PEG 400, as an example of very highly viscous solvents, could need about one day to dissolve PLGA at a concentration of only 10%. This time could be more consuming along with increasing the polymer concentration due to the increased viscosity of the prepared formulation.

From this perspective, the viscosity of solvents is a crucial factor while preparing the ISGI systems. To select a proper organic solvent, it should have a suitable low viscosity as the lower the viscosity value of the solvent, the easier the injection of ISGI preparation from syringe needles [69,70]. Using solvents with high affinity to the polymer can promote dissolution and decrease the overall viscosity of the prepared formulations. In addition, low volumes of solvents required to dissolve the polymer can be utilized, therefore adverse effects of metabolic products of organic solvents can be obviated [71]. In contrast, solvents with low affinity to the polymer can certainly lead to the preparation of highly viscous formulations with the possible appearance of polymeric aggregates. This can complicate the injection of the formulation from needles and produce painful and difficult administration to patients resulting in non-compliance of patients [72,73]. These explanations could support our preliminary study results, and hence NMP, BA, DMSO, and EA were selected for further investigation. 

After selecting the appropriate solvents, alogliptin was loaded in the corresponding PLGA-solvent mixtures for preparing the drug-loaded ISGI formulations. These were composed of PLGA (85:15) dissolved in NMP, BA, DMSO, or EA solvents at a constant solvent to polymer ratio of 2:1. The in vitro release profile of alogliptin from the prepared ISGI formulations was studied in comparison to that of the pure drug solution as represented in Figure 1. The pure alogliptin solution showed a rapid initial burst release in the first 6 h (73.92 ± 0.25%) with maximum release after only one day (99.32 ± 1.71%). In comparison, the four studied alogliptin-loaded ISGI formulations displayed more sustained release profiles of the drug. The ISGI formulations containing BA and EA demonstrated higher initial burst release of the drug after 6 h (41.53 ± 0.19% and 28.06 ± 1.29%, respectively) with sustained drug release after 10 days (65.26 ± 0.72% and 35.72 ± 0.55%, respectively). However, the ISGI formulations containing NMP and DMSO solvents showed a remarkable decrease in the initial burst release of alogliptin in the first 6 h (3.28 ± 1.04% and 5.30 ± 0.25%, respectively) and presented a much better-sustained release profile till reaching the tenth day of release study (43.08 ± 1.73% and 51.92 ± 0.58%, respectively).

The initial burst release, as the first stage of the release profile of drugs from ISGI systems, is certainly fundamental to be studied after preparing these systems [74]. The presence of burst release effect after minutes or hours after injecting the ISGI preparations can exist due to several mechanisms. For illustration, a lag time may occur between injecting the liquid polymeric solution and its solidification at the site of injection during the solvent exchange process [75]. Besides, both solvent and dissolved drug may rapidly transit out from the ISGI formulation resulting in rapid initial burst release of drug [76]. Moreover, free drugs may be present on the outer surface of solidified ISGI system or may be irregularly distributed inside the solidified ISGI system leading to a rapid burst release [77]. All of these causes may result in inconvenient drug loading with the possible incidence of local or systemic adverse effects after injection into the body. Therefore, the initial burst effect is a challenging phenomenon that should be taken into consideration while preparing the ISGI systems.

In our study, the burst release of alogliptin was influenced according to the type of solvent where ISGI formulations containing BA and EA showed a more rapid burst release of drug than those containing NMP and DMSO. This might be ascribed to the low miscibility of BA and EA solvents in water compared to the higher water miscibility of NMP and DMSO solvents. Hence, a lag time was required for the former solvents between injecting the preparation into the buffer release medium and solidification of ISGI formulations leading to a marked fast initial burst release of alogliptin. These findings were in agreement with that stated by Kilicarslan et al. [78] and Turino et al. [79] where the utilization of low water-miscible solvents could cause no rapid precipitation of the used polymer followed by slow hardening of the injected preparation and incidence of fast initial burst effect. However, the latter highly water-miscible solvents could provide more homogenous gel preparations with low viscosity features that facilitated faster phase inversion dynamics within seconds or minutes during the solvent exchange stage [71]. This could create a highly hydrophilic medium where a rapid influx of buffer into the polymeric solution and a rapid outflux of such solvents could be provided resulting in rapid solidification of ISGI systems having highly entrapped drug amount and lower initial burst release of the drug [36]. Accordingly, BA and EA solvents were excluded from further evaluations due to their unsatisfactory results in controlling the initial burst release of alogliptin from the formulated ISGI systems. Meanwhile, good injectable NMP and DMSO solvents were selected to study their effects on the burst and cumulative release of alogliptin along with changing other factors using a computer-based full factorial design.

### 3.2. Statistical Data Analysis

A total of sixteen runs suggested by the factorial design were prepared and their responses were evaluated according to changing the independent factors within two levels. The independent factors were lactide concentration in PLGA (A), type of solvent (B), and PLGA amount (C). The dependent responses to be measured were burst release of alogliptin after 6 h (Y_1_) and cumulative release of alogliptin after 10 days (Y_2_). The observed responses of the formulations were analyzed using Design-Expert^®^ software. The polynomial equations that described the individual main effects and interaction effects showed a significant influence on the studied responses. Table 4 represents the good correlation between the actual and predicted values of Y_1_ and Y_2_ responses.

#### 3.2.1. Effect of Independent Factors on Y_1_ Response

The results of the factorial design were firstly analyzed using a half-normal plot and Pareto chart to indicate the order of significant effects on the resultant responses. In the half-normal plot, the higher significant factors appeared on the upper right side and the smaller significant effects or noise appeared on the lower left side [80]. The Pareto chart is a graphical presentation that shows the significance of independent variables using ordered bars, where the t value of each effect was checked by two limit lines (Bonferroni and t value limit lines) [81]. The t value of the effect above the Bonferroni limit line confirmed the high significance of such an effect. When the t value was found between the two limit lines, the effect was likely to be significant. However, the effect having a t value below the limit lines was considered statistically non-significant [82]. Herein, increasing the PLGA amount (factor C) could significantly decrease the initial burst release of alogliptin rather than the other lower significant factors (B and A, respectively), as shown in Appendix A. Moreover, the % contribution of each factor to the initial burst release of alogliptin (Y_1_ response) could signpost how the factor could participate and contribute to influencing the measured response [47]. As shown in Table 5, factor C possessed the highest impact on the Y_1_ response (89.72%) compared with the other factors. Although the lactide concentration in PLGA and solvent type (A and B factors, respectively) could significantly affect the initial drug burst release (*p* < 0.0001), they influenced the Y_1_ response with a smaller % contribution (2.67 and 4.22%, respectively). 

The results of ANOVA statistical analysis of the Y_1_ response (Table 5) represented that the model was significant as known through the high F-value (415.84) of *p*-value < 0.0001. This meant that there was only a 0.01% chance of the presence of this high F-value due to noise. By monitoring the *p*-values presented in Table 5, the A, B, C, AC, and BC terms of *p*-values less than 0.05 were significant model terms while other model terms were not significant. The R^2^, a measure of the amount of variation around the mean explained by the model, was found to be high (0.9973) and close to 1. This could reflect the good correlation between the actual and predicted values of the Y_1_ response [83]. The adjusted R^2^ could assess the amount of variation around the mean adjusted for the number of terms in the model. The predicted R^2^ could express how the model could predict responses for new observations [84]. The predicted R^2^ (0.9890) and adjusted R^2^ (0.9949) were in good agreement with each other since the difference between them was less than 0.2 as required by the design software. In addition, the C.V.%, the ratio of the standard deviation to the mean, was found to have a small value (5.83), indicating the lower level of dispersion around the mean and good relation between the experimental values and those of the fitted models [85]. Adequate precision could compare the range of the predicted values to the average prediction error and its value was required to be higher than 4 for ensuring the adequate model discrimination [86]. This value was 50.7598 which expressed an adequate signal/noise ratio and that the model could navigate the design space. The PRESS could indicate how the model could fit each point in the used design and its low value (95.78) could estimate the significant fitting of the model [87].

The polynomial equation for this model was determined as:Y_1_ = 29.68 − 3.82 A − 4.80 B − 22.13 C + 0.17 AB + 2.45 AC + 3.30 BC − 0.29 ABC

The polynomial equation was obtained to predict the response using given levels of each factor in the design. The coefficients of each factor in the equation could be beneficial for recognizing the relative impact of factors on the studied responses [85]. As reported by Hosny et al. [88], the positive sign of the factor coefficient could indicate the synergistic effect while the negative sign could point out the antagonistic effect of such a factor on the measured response. The aforementioned equation could reveal the negative impact of increasing the lactide concentration and the amount of PLGA polymer on the initial burst release of alogliptin from the prepared ISGI formulations.

By analyzing the model diagnostic plots, the normal plot of residuals could indicate whether the difference between actual and predicted values of the Y_1_ response could follow a normal distribution and thus follow a straight line [89]. As shown in Appendix A, the normal distribution of error could be seen as the residual values were predominantly fitted on a straight line. Besides, the externally studentized residuals versus runs plot was constructed to assure the desirable suitability and fitness of the utilized model, where a random trend was monitored (Appendix A) where the data points were located within the range of control limit lines. This could suggest the development of experiments in a random manner showing satisfactory fit with minimization of errors [53].

To visualize the influence of the independent individual/interaction factors on the measured response (Y_1_), the model graphs such as perturbation, one factor, and interaction plots were obtained. The perturbation plot could offer a comparison between the effects of all independent variables at a reference point in the design space where the studied response could be plotted by changing factors over their ranges and keeping the other factors constant [59,90]. The perturbation plots using DMSO solvent (Appendix A) or NMP solvent (Appendix A) showed that increasing the PLGA amount could be more required to decrease the initial burst release than other independent variables.

Through the one-factor plots, the effects of factors were plotted where two factors were kept constant and the other factor was studied within its high and low levels. Hence, the action of the latter factor could be seen as a line displaying its effect on the demanded response [91]. Figure 2a shows that the initial burst release of alogliptin could be decreased from 38.47% to 30.49% by increasing the lactide concentration to 85% with the utilization of DMSO solvent and the amount of polymer in its medium level (250 mg). The increment of lactide concentration could also minimize the Y_1_ response from 28.53 to 21.23% using NMP solvent and the amount of PLGA in its medium level (Figure 2b). Therefore, studying the influence of lactide concentration in the PLGA polymer on controlling the burst drug release from ISGI formulations was certainly fundamental. Different distributions of lactide and glycolide in the PLGA structure could modulate the hydrophobicity of the ISGI system controlling the water uptake and then the rate of system degradation [92]. By comparing the ISGI formulations with the same components but different lactide concentrations (Table 4), the initial burst release of alogliptin could be orderly minimized, for example, F1 (6.28%) > F5 (3.29%), F9 (9.19%) > F2 (5.30%), F7 (49.04%) > F11 (37.26%), and F6 (66.15%) > F4 (52.55%) (Figure 3a). The higher burst release of alogliptin was exhibited by ISGI formulations with lower concentrations of lactide (65%) compared to others with higher proportions of lactide (85%). These findings could be owing to the higher lipophilic nature of lactide than glycolide leading to decreased water uptake, acceleration of ISGI solidification and augmentation of alogliptin entrapment in the ISGI systems while increasing the concentration of lactide in the PLGA polymer [93].

On the other hand, the burst release of the drug was reduced from 34.48% to 24.88% by replacing DMSO with NMP while keeping both factors A and C at their medium levels (Figure 2c). The effects of the nature and characteristics of solvents used for preparing the ISGI systems in terms of water miscibility, solvation power and their affinity toward the utilized polymer were necessary to be studied where using different solvents could induce a strong impact on the burst drug release [20]. Although both NMP and DMSO solvents used in our study were water-miscible possessing fast phase inversion kinetics, they displayed a relative difference towards their potential on the initial burst release of alogliptin after 6 h. As shown in Table 4 and Figure 3b, the Y_1_ response was decreased in the following order: F13 (11.39%) > F3 (8.18%), F10 (10.30%) > F15 (5.82%), F8 (67.14%) > F14 (49.99%), and F12 (53.84%) > F16 (38.56%). The burst drug release was reduced by the ISGI formulations containing NMP solvent in comparison to those containing DMSO solvent. This could be ascribed to the variations in the solvating potential of both solvents as stated by Ahmed et al. [25] where the water miscibility of DMSO was higher than that of NMP. Hence, DMSO solvent could display higher affinity to water than to PLGA polymer, producing faster phase inversion during the solvent exchange step. Then, some of the alogliptin could dissipate into the release medium in a rapid manner with a higher initial burst release. Furthermore, Hildebrand’s solubility parameter (δ) could be used to assess the affinity between the solvent and the solute where those having similar δ values could dissolve easily in each other showing high mutual affinity [65]. Following equations of calculating the δ parameters of DMSO, NMP, and PLGA as reported by Zhang et al. [94], the close δ parameter of PLGA (22.30 (cal/cm^3^)^0.5^) to that of NMP (22.90 (cal/cm^3^)^0.5^) rather than that of DMSO (19.40 (cal/cm^3^)^0.5^) in addition to the lower Δδ of NMP/PLGA (4.80 MPa^0.5^) than that of DMSO/PLGA (7.60 MPa^0.5^), could support our findings about the lower affinity of DMSO solvent to PLGA than that of NMP to PLGA which could correspond to faster DMSO exchange rates and higher initial burst drug release. Moreover, the increment of solvent affinity to polymer could raise the viscosity of the prepared ISGI system due to the higher polymer interactions that occurred as a result of decreasing the rate of drug diffusion [93].

Concerning the most effective variable (factor C) at the medium level of factor A (75%), the measured Y_1_ response was steeply decreased from 59.92% to 9.05% (Figure 2d) and from 43.71% to 6.05% (Figure 2e) using DMSO and NMP solvents, respectively. The impact of changing PLGA amount was clearly shown by the large gradient of the slope. As per the data in Table 4 and Figure 3a, the Y_1_ response was markedly reduced by increasing the PLGA amount as shown in the following sequence: F7 (49.04%) > F1 (6.28%), F11 (37.26%) > F5 (3.29%), F6 (66.15%) > F9 (9.19%), and F4 (52.55%) > F2 (5.30%). The formulations with a higher amount of PLGA represented a significantly smaller initial burst release than that produced by formulations containing a lower amount of PLGA polymer. This might be attributed to the fact that the increment of the thickness of polymeric skin could be exhibited by increasing the polymer concentration, leading to retardation of solvent/water exchange and the production of less porous formulations [26]. In addition, decreasing the solvent amount during the ISGI preparation by increasing the PLGA amount could increase the viscosity of the prepared systems, resulting in reducing the initial release of the drug into the external buffer [74]. 

The interaction plots could illustrate how the interaction of various factors could significantly impact the studied response depending on changing the levels of two factors. Non-parallel or crossed lines could indicate that one factor could rely on the level of another factor. However, parallelism could elucidate the lack of interaction between the studied two factors [95]. As shown in Figure 4a, the initial burst release of alogliptin after 6 h could be insignificantly decreased using either DMSO or NMP solvents while increasing the lactide concentration to 85% and using a PLGA amount of 250 mg. Thus, there was no interaction effect between the two factors (A and B) on the Y_1_ response as elicited from the two parallel lines that appeared in Figure 4a. This was consistent with the results of Table 5 where the interaction term AB had no significant effect with a *p*-value of 0.7094. However, there was a significant decrease in the initial drug release (7.80%) by increasing factors (A and C) to their high levels and using DMSO as a solvent compared to the smaller decrease in burst release to 53.19% following the same conditions, but using the PLGA polymer at its lowest level (Figure 4b). The burst release could be significantly reduced to 4.55% by dissolving PLGA with 85% lactide in NMP solvent at a ratio of 1:2 in comparison to that reduced to 37.91% using PLGA with 85% lactide and NMP at a ratio of 1:4 (Figure 4c). Hence, the interaction effect of the A and C factors was remarkably observed. In addition, the interaction plot of variables (B and C) displayed the highly significant impact of changing the PLGA amount from its lowest to its highest level on the burst release as clarified by the remarkable distance between the upper black line and lower red line, where the former line expressed the lowest level of factor C and the latter one expressed the highest level of such factor (Figure 4d). Therefore, the influence of alteration of PLGA amount in the ISGI formulation was evident to significantly influence the initial burst release of alogliptin after 6 h rather than the other two studied variables and this was in a feasible agreement with the steeper slope of factor C as seen in the perturbation and one-factor plots.

#### 3.2.2. Effect of Independent Factors on Y_2_ Response

According to analyzing the impact of studied variables on the cumulative release of alogliptin after 10 days, a half-normal plot and Pareto chart were firstly followed to determine how the factors were arranged corresponding to their significant actions on the Y_2_ response. Appendix A represented the higher significant decrease in the cumulative release of alogliptin by the action of variable C rather than the lower significant action of A, B, and BC factors, respectively. In addition, the % contribution of each variable to the Y_2_ response was examined where the variable of PLGA amount demonstrated the highest negative impact on the cumulative alogliptin release with a % contribution of 85.26% compared to other variables (Table 6).

The ANOVA statistical analysis (Table 6) showed the significance of the Y_2_ model owing to its high F-value (167.04) and low *p*-value (*p* < 0.0001). This meant that there was only a 0.01% chance of the presence of this high F-value due to noise. Besides, the A, B, C, and BC terms of *p*-values less than 0.05 were recognized as significant model terms, while other model terms were considered not significant. The predicted R^2^ (0.9728) and adjusted R^2^ (0.9873) were in feasible agreement with each other since the difference between them was less than 0.2 as required by the design software. The C.V.% was found to be small (2.20), indicating the lower dispersibility of the values around the mean. The adequate precision value was greater than 4 (35.3427), indicating an adequate signal/noise ratio and that the model could express the design space. The low value of PRESS (61.51) could assess the significant fitting of the model. The polynomial equation for this model was determined as:Y_2_ = 62.97 − 3.81 A − 1.74 B − 10.98 C − 0.67 AB + 0.41 AC − 1.05 BC − 0.79 ABC

The equation could manifest the negative influence of increasing the lactide concentration and the amount of PLGA polymer on the cumulative release of alogliptin after 10 days.

Regarding the model diagnostic plots, the normal distribution of residuals (Appendix A) could display the residuals values as close to the straight line to a major extent. Furthermore, the externally studentized residuals versus runs plot was constructed showing a random trend and location of data points between the control limit lines (Appendix A). This could emphasize the desirable suitability and fitness of the model with lower error chances. Moreover, the perturbation plots were performed where the alogliptin release from the ISGI formulations was highly sensitive to the alterations of solvent to PLGA ratios using DMSO solvent (Appendix A) or NMP solvent (Appendix A).

By studying the one-factor plots, Figure 5a showed that the cumulative release of alogliptin could be reduced from 67.85% to 61.57% by increasing the lactide concentration to 85% with the utilization of DMSO solvent and the amount of PLGA in its medium level (250 mg). In addition, the increment of lactide concentration could also decrease the Y_2_ response from 65.71% to 56.75% using NMP solvent and the amount of PLGA at its medium level (Figure 5b). Thus, it was noteworthy to study the influence of lactide concentration in the PLGA polymer on controlling the cumulative release of alogliptin from the ISGI formulations. The results of Table 4 mentioned that the Y_2_ response was decreased in the following order: F1 (53.14%) > F5 (43.08%), F9 (55.59%) > F2 (51.92%), F7 (76.32%) > F11 (68.17%), and F6 (78.04%) > F4 (69.35%) (Figure 3a). The higher cumulative release of alogliptin was observed in the ISGI formulations with lower concentrations of lactide (65%) compared to others with higher proportions of lactide (85%). The variations in the in vitro profile of alogliptin-loaded ISGI systems could exist by changing the chemical composition of PLGA polymer where changing the lactide/glycolide ratios could change the physical characteristics of the ISGI matrix, such as hydrophobicity, crystallinity, and permeability. The increment of lactide concentration could facilitate decreasing the water absorption and decelerating the drug release [28,96].

On another side, the cumulative release of alogliptin could be minimized from 64.71% to 61.23% by replacing DMSO with NMP solvent while keeping both factors A and C at their medium levels (Figure 5c). As displayed in Table 4 and Figure 3b, the order of decreasing the Y_2_ response could be represented as follows: F13 (57.44%) > F3 (54.99%), F10 (53.77%) > F15 (45.58%), F8 (79.91%) > F14 (78.39%), and F12 (71.24%) > F16 (70.19%). The cumulative drug release was reduced by the ISGI formulations containing NMP solvent in comparison to those containing DMSO solvent. This could be attributed to the higher affinity of NMP to PLGA rather than DMSO which could increase the viscosity of the prepared ISGI system containing NMP. This could participate in decreasing the diffusion of alogliptin particles with the appearance of slower in vitro drug release profiles [38,72,93]. 

Regarding the analysis of factor C at the medium level of factor A (75%), the measured cumulative release after 10 days was steeply decreased from 74.64% to 54.68% (Figure 5d) and from 73.26% to 49.19% (Figure 5e) using DMSO and NMP solvents, respectively. The influence of increasing factor C was clearly shown by the steep slope of the corresponding one-factor plots. As per results of Table 4 and Figure 3a, the Y_2_ response was obviously reduced by increasing the PLGA amount as shown in the following sequence: F7 (76.32%) > F1 (53.14%), F11 (68.17%) > F5 (43.08%), F6 (78.04%) > F9 (55.99%), and F4 (69.35%) > F2 (51.92%). The formulations with a higher amount of PLGA presented a significantly smaller cumulative release of alogliptin after 10 days than that produced by formulations containing a lower amount of PLGA polymer. This could be ascribed to the fact that increasing the PLGA polymer molecules in comparison to drug molecules could facilitate the ability of PLGA to encapsulate more molecules of alogliptin, resulting in a lower proportion of drugs going out to the external buffer with the achievement of a more sustained release profile of drug [97]. Furthermore, increment of PLGA amount resulted in utilization of lesser amounts of solvents which could increase the polymer-polymer interactions with increased overall viscosity of the prepared ISGI systems. Thus, water uptake and rate of drug diffusion could be diminished with the incidence of a sustained cumulative drug release [24,71].

For studying the interaction of various factors and their impacts on the studied Y_2_ response, the interaction plots were used depending on changing the levels of two factors together. As shown in Figure 6a–c, the two lines were parallel to some extent indicating the lack of interaction effects between the studied two factors and each factor showed its individual effect on the Y_2_ response [98]. This was in an agreement with the results of Table 6 where the interaction terms (AB and AC) showed no significant interaction effects with *p*-values of 0.0892 and 0.2751, respectively. However, Figure 6d displayed the significant interaction effect of factors (B and C) on the Y_2_ response. The cumulative release of alogliptin could be significantly decreased using higher levels of PLGA dissolved in the NMP solvent, while keeping the lactide concentration at 75%, for preparing the ISGI systems rather than using DMSO solvent.

### 3.3. Optimization Process and Statistical Validation

The optimization was carried out for obtaining desirable levels of studied variables corresponding to the goals needed for the measured responses as shown in Table 1. The required criteria relied on minimizing the initial burst release of alogliptin from the ISGI systems after 6 h and increasing its cumulative release after 10 days. Concerning the highest desirability value (0.747), the formulation containing PLGA polymer (312.5 mg) with a lactide concentration of 65% and DMSO solvent was selected as an optimized formulation. As given by the factorial design software, the predicted values for Y_1_ and Y_2_ responses were 10.29% and 56.72%, respectively. The optimized formulation was prepared and subjected to in vitro release study where the experimental values for Y_1_ and Y_2_ responses were found to be 9.89% and 53.93%, respectively (Figure 7). The observed and predicted values were in good agreement with % prediction error values of less than 5%. This could indicate the closeness of the observed and predicted values and the adequate ability of optimization tools in the study. Hence, the validity, reliability, and applicability of the models to appropriately describe the influence of studied factors on the measured responses were confirmed [50,54].

For determining the mechanism of alogliptin release from the optimized ISGI formulation, the in vitro release profile was examined using various mathematical models namely; zero order, first order, Higuchi, Hixson–Crowell, and Korsmeyer–Peppas models as shown in Table 7. The model that presented higher R^2^ could best describe the mechanism of drug release. When compared to other tested models, the alogliptin-loaded ISGI formulation exhibited a release profile best fitted to the Higuchi model equation (R^2^ = 0.9807), which reflected the diffusion behavior of alogliptin release [27]. In addition, the drug release mechanism was subjected to Korsmeyer–Peppas model where the n value was 0.445 (less than 0.5). This could express the fitness of drug release to the Fickian diffusion mechanism (case I transport) [70].

### 3.4. Characterization of Optimized ISGI Formulation

#### 3.4.1. SEM Study

The morphology of the optimized ISGI formulation including the internal and external structure was examined using a scanning electron microscope. After being injected into the buffer, the formulation was withdrawn after 0, 1, and 10 days to examine its morphology. Immediately after injection at 0 days, the external surface (Figure 8a) and internal structure (Figure 8b) of the optimized preparation did not show any pores exhibiting a homogeneous, uniform, and smooth appearance. After 1 day, the external surface presented some fissures and incisions which might reflect the increment of the thickness of the external shell of the formulation (Figure 8c). This could be due to the higher affinity of DMSO solvent to water producing a faster phase inversion and higher water uptake during the solvent exchange step leading to the overall expansion of the formulation [19,99]. Besides, the internal structure of the ISGI formulation exhibited a slightly spongy appearance with the presence of some holes (Figure 8d), which could explain the release of alogliptin from the formulation [100]. On another side, the preparation surface became highly wrinkled and irregular along with the appearance of pores after 10 days (Figure 8e). This could reveal the onset of polymer degradation and the release of more amounts of alogliptin into the external medium [101]. Furthermore, the cross-section displayed an increasing number and enlargement of internal holes (Figure 8f). Fialho and Cunha [102] reported that the degradation of PLGA polymer could occur faster from the inside than from the outside of the matrix with the expansion of internal water channels that could enhance the drug diffusion from the implant preparation. 

#### 3.4.2. FTIR Study

The FTIR study was conducted to investigate the chemical interactions between the drug and the components used in the preparation of the ISGI system by monitoring any shift or disappearance of spectra peaks as demonstrated in Figure 9. The spectrum of pure alogliptin showed characteristic absorption bands at 3447 cm^−1^ (NH_2_), 3083 cm^−1^ (aromatic C-H), 2957, 2858 cm^−1^ (aliphatic C-H), 2229 cm^−1^ (C≡N), 1697 cm^−1^ (C=O), 1618 cm^−1^ (N-H bending), 1542 cm^−1^ (aromatic C = C), 1448 cm^−1^ (C-H bending), and 1212 cm^−1^ (C-N). The PLGA spectrum displayed peaks at 3483 cm^−1^ (O-H), 2998, 2952 cm^−1^ (aliphatic C-H), 1756 cm^−1^ (C=O), 1457 cm^−1^ (C-H bending), and 1092 cm^−1^ (C-O). The physical mixture presented no chemical interactions between alogliptin and PLGA polymer when they were used together in a solid state. However, some changes in the FTIR spectrum appeared by dissolving the PLGA in DMSO solvent. As seen in the spectrum of the blank ISGI formulation, a distinct broad peak appeared at 3419 cm^−1^ and sharp peaks presented at 1757 and 1022 cm^−1^. These alterations might be due to the hydrogen bonds between the free O-H group of PLGA and the sulfoxide group of solvent used, whereas the optimized ISGI formulation showed an absence of characteristic peaks of alogliptin with the formation of no new peaks in the FTIR spectrum. This indicated an absence of chemical interactions between the drug and excipients along with the presence of the drug in its amorphous form in the prepared ISGI formulation. These findings were in agreement with results reported by Hiremath et al. [100] and Ibrahim et al. [103].

#### 3.4.3. DSC Study

The chemical interactions between drug and excipients were determined by DSC analysis where any shifting or disappearance of peaks could indicate the alteration in the drug crystallinity. As presented in Figure 10, pure alogliptin showed a sharp endothermic peak at 185.71 °C corresponding to its crystalline state at this melting point. The PLGA polymer demonstrated a characteristic peak at 45.43 °C corresponding to its glass transition temperature [104]. The physical mixture exhibited no chemical interaction between alogliptin and PLGA polymer upon using both components in a solid state. Meanwhile, the thermal behavior of the blank ISGI formulation demonstrated the possible interactions between PLGA and DMSO, as seen by the disappearance of the glass transition temperature of the polymer. Likewise, no difference was observed in the thermal behavior of the optimized ISGI formulation compared to that of the blank formulation and the characteristic peak of alogliptin did not appear. This revealed the transformation of crystalline alogliptin to an amorphous state while being incorporated into the ISGI system.

#### 3.4.4. Influence of Oral Alogliptin Solution and Subcutaneous Optimized ISGI Formulation on Blood Glucose Level of Rats

Measurement of postprandial hyperglycemia can be generally considered one of the more sensitive indicators of diabetic control. It can rapidly reflect the hypoglycemic effect induced by the drug’s action on the blood glucose levels [105]. As shown in Figure 11, the diabetic animals (group 2) showed significant clear hyperglycemia profiles as shown by the marked stable and high postprandial blood glucose levels following the induction of diabetes in comparison with the corresponding time points of the control rats (group 1) at 1, 3, 7, and 10 days (all at *p* < 0.05). The daily administration of oral alogliptin (2.5 mg/kg) demonstrated an improvement in the glycemic control of the hyperglycemia-induced rats (group 3) and this could be seen by the significant decrement in the blood glucose levels at the 1, 3, 7, and 10 days after induction of diabetes (all at *p* < 0.05) in comparison with those of the diabetic rats (group 2).

On another side, a single subcutaneous injection of the optimized alogliptin-loaded ISGI preparation (25 mg/kg) produced even more improvement in the glycemic control of diabetic animals compared with that produced by the daily administration of oral alogliptin preparations at all time points. The improvement in the glycemic control shown by the single subcutaneous injection of the optimized alogliptin-loaded ISGI preparation was more significantly effective than those produced by the frequent daily administration of the oral alogliptin preparations, especially at the 3, 7, and 10 days after first administration (*p* < 0.05). This could be observed by the significant decrease in the blood glucose levels of diabetic animals shown throughout the study period when compared with those of group 2 animals (*p* < 0.05). These results were due to the sustained release of alogliptin from the optimized ISGI formulation over extended time periods, as shown by the in vitro release tests. Our findings revealed the reflective potential of the injectable ISGI preparation loaded with anti-hyperglycemic alogliptin to effectively minimize the blood glucose levels for longer durations with the achievement of possible safe administration of an optimized depot alogliptin-ISGI dose. This could improve patient compliance and diminish the side effects induced by the frequent doses of the drug with no risks of inadvertent hypoglycemia.

## 4. Conclusions

In this study, the PLGA polymer was used to develop ISGI systems loaded with anti-diabetic alogliptin using different organic solvents. Such solvents were selected depending on their physicochemical characteristics, such as solubilization capacity, viscosity, water miscibility, and ability to dissolve the polymer. Following the 2^3^ factorial design, the negative influence of increasing the lactide concentration and the amount of PLGA on the initial burst and cumulative release of alogliptin was shown. The experimental values for Y_1_ and Y_2_ responses of the optimized ISGI formulation were found to be 9.89% and 53.93%, respectively with a desirability value of 0.747. When subcutaneously injected into diabetic rats, the optimized ISGI preparation displayed a significant decrease in the animals’ blood glucose levels throughout the study period compared to that of the oral alogliptin solution. Our study signified the benefits of the injectable long-acting alogliptin-loaded ISGI systems for treating type 2 diabetes by enhancing the compliance and adherence of patients. For future perspectives, more innovative strategies should be developed for incorporating multiple drugs into the PLGA-ISGI systems to be available in the market for achieving multi-target drug approaches. 

## Figures and Tables

**Figure 1 pharmaceutics-14-01867-f001:**
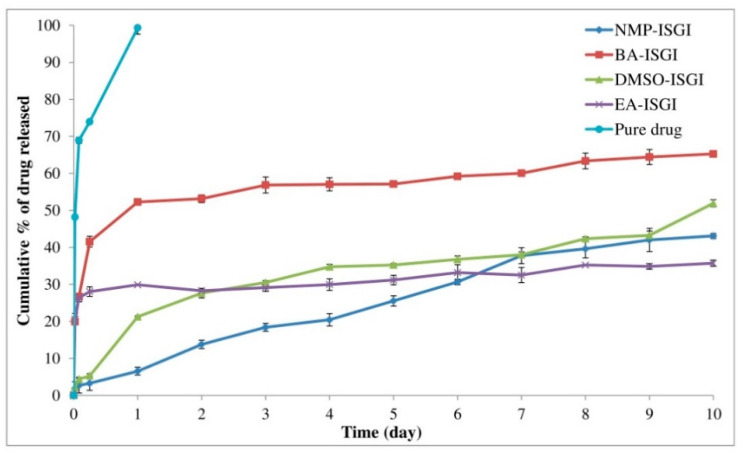
Preliminary study of in vitro release of alogliptin-loaded ISGI formulations containing different solvents.

**Figure 2 pharmaceutics-14-01867-f002:**
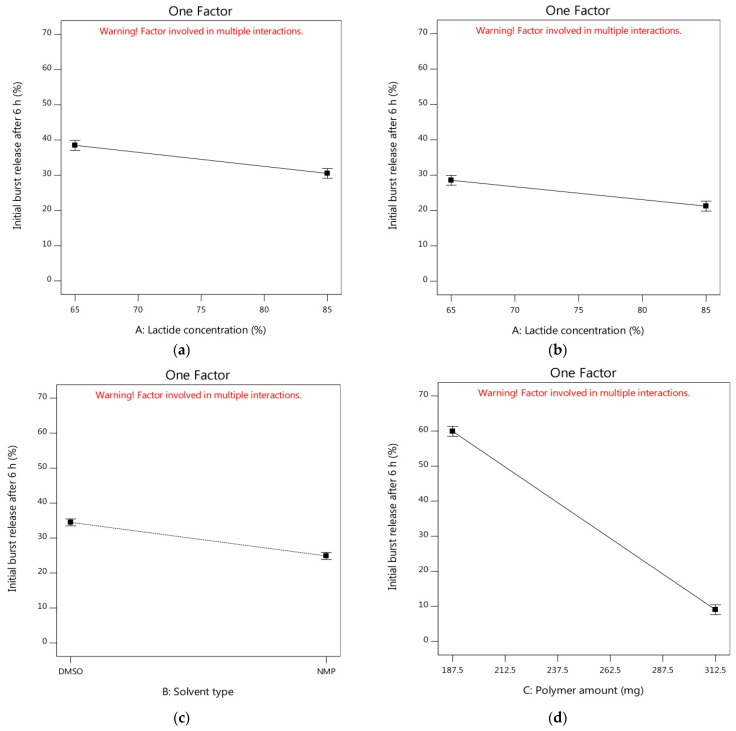
One factor plot of influence of independent factors on Y_1_ response (**a**) effect of A using DMSO and medium level of C (**b**) effect of A using NMP and medium level of C (**c**) effect of B using medium levels of A and C (**d**) effect of C using DMSO and medium level of A (**e**) effect of C using NMP and medium level of A.

**Figure 3 pharmaceutics-14-01867-f003:**
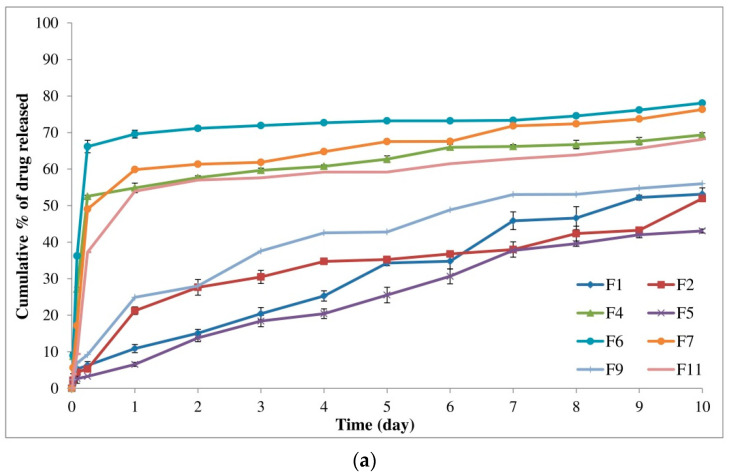
In vitro release of alogliptin-loaded ISGI formulations showing (**a**) individual effects of A and C on measured responses (Y_1_ and Y_2_) (**b**) individual effect of B on measured responses (Y_1_ and Y_2_).

**Figure 4 pharmaceutics-14-01867-f004:**
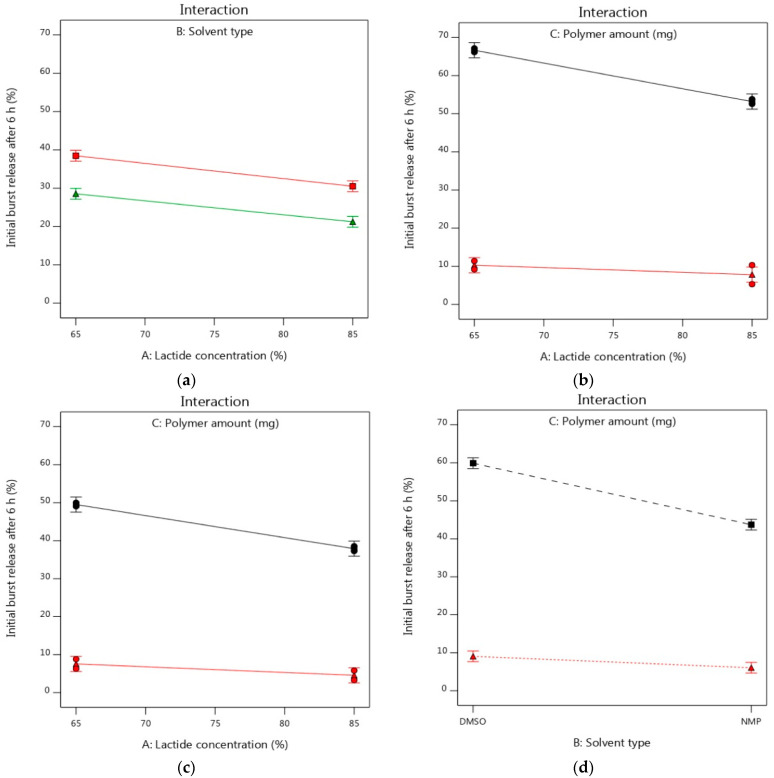
Interaction plots of influence of independent factors on Y_1_ response (**a**) effects of A and B (**b**) effects A and C using DMSO (**c**) effects of A and C using NMP (**d**) effects of B and C. Figure 4a: the upper line refers to DMSO and the lower line refers to NMP. Figure 4b–d: the upper black lines refer to PLGA (187.5 mg) and the lower red lines refer to PLGA (312.5 mg).

**Figure 5 pharmaceutics-14-01867-f005:**
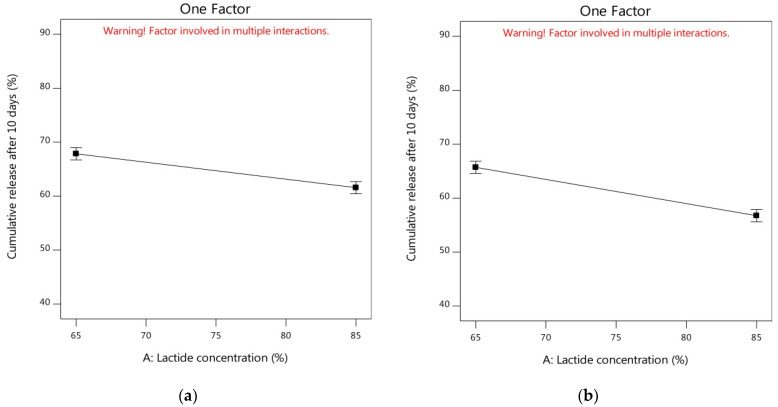
One factor plots of influence of independent factors on Y_2_ response (**a**) effect of A using DMSO and medium level of C (**b**) effect of A using NMP and medium level of C (**c**) effect of B using medium levels of A and C (**d**) effect of C using DMSO and medium level of A (**e**) effect of C using NMP and medium level of A.

**Figure 6 pharmaceutics-14-01867-f006:**
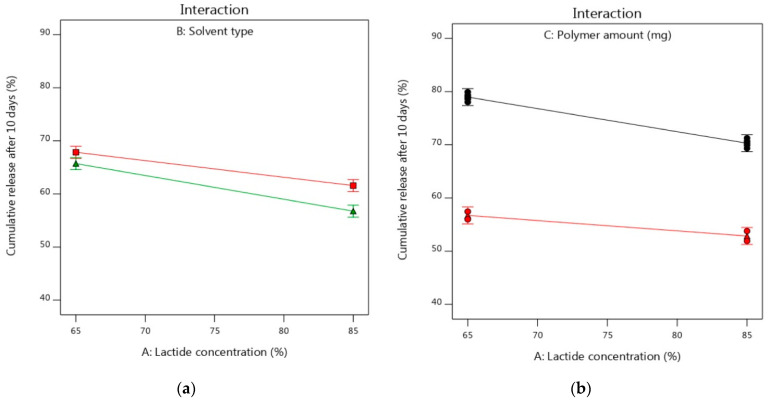
Interaction plots of influence of independent factors on Y_2_ response (**a**) effects of A and B (**b**) effects A and C using DMSO (**c**) effects of A and C using NMP (**d**) effects of B and C. Figure 5a: the upper line refers to DMSO and the lower line refers to NMP. Figure 5b–d: the upper black lines refer to PLGA (187.5 mg) and the lower red lines refer to PLGA (312.5 mg).

**Figure 7 pharmaceutics-14-01867-f007:**
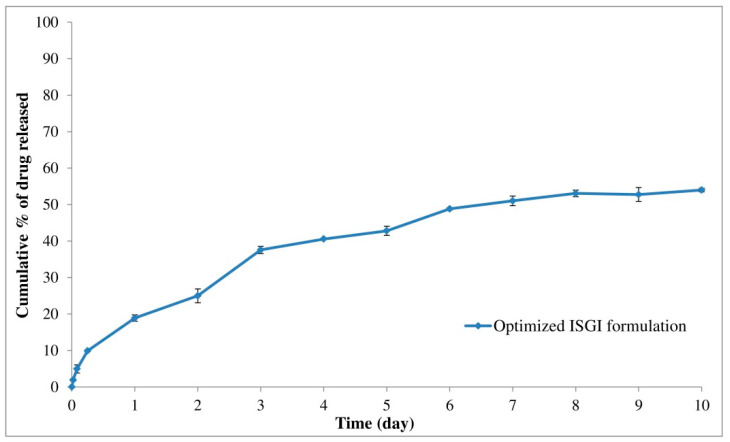
In vitro release of optimized alogliptin-loaded ISGI formulation.

**Figure 8 pharmaceutics-14-01867-f008:**
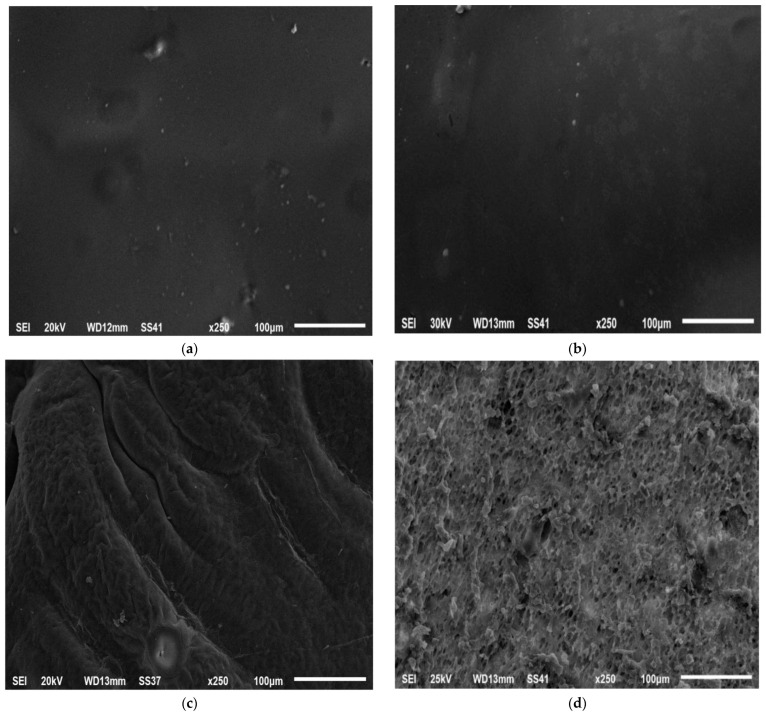
SEM study of optimized ISGI formulation (**a**) external surface after 0 day (**b**) internal structure after 0 day (**c**) external surface after 1 day (**d**) internal structure after 1 day (**e**) external surface after 10 days (**f**) internal structure after 10 days.

**Figure 9 pharmaceutics-14-01867-f009:**
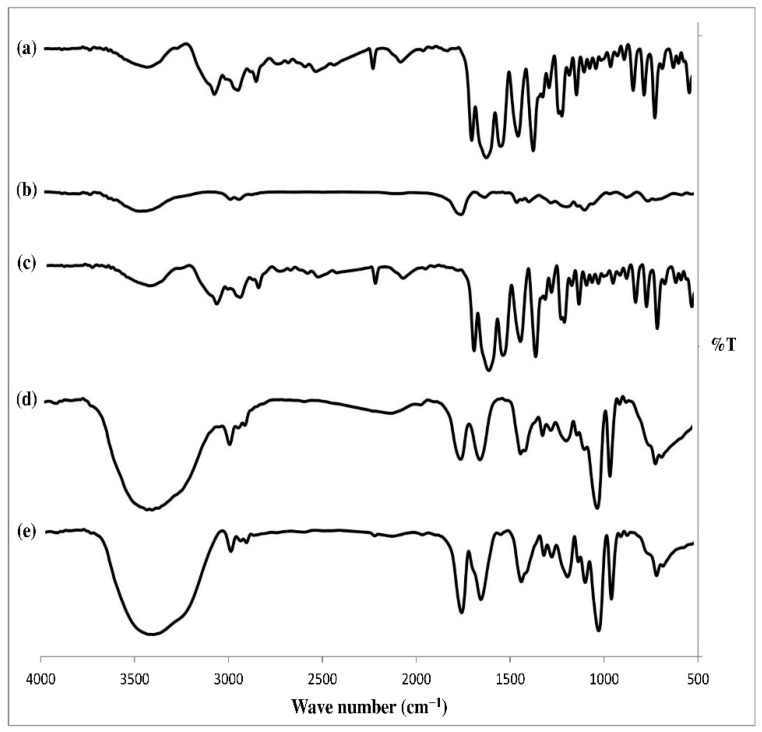
FTIR spectrums of (**a**) pure alogliptin (**b**) PLGA (**c**) physical mixture (**d**) blank ISGI formulation (**e**) optimized ISGI formulation.

**Figure 10 pharmaceutics-14-01867-f010:**
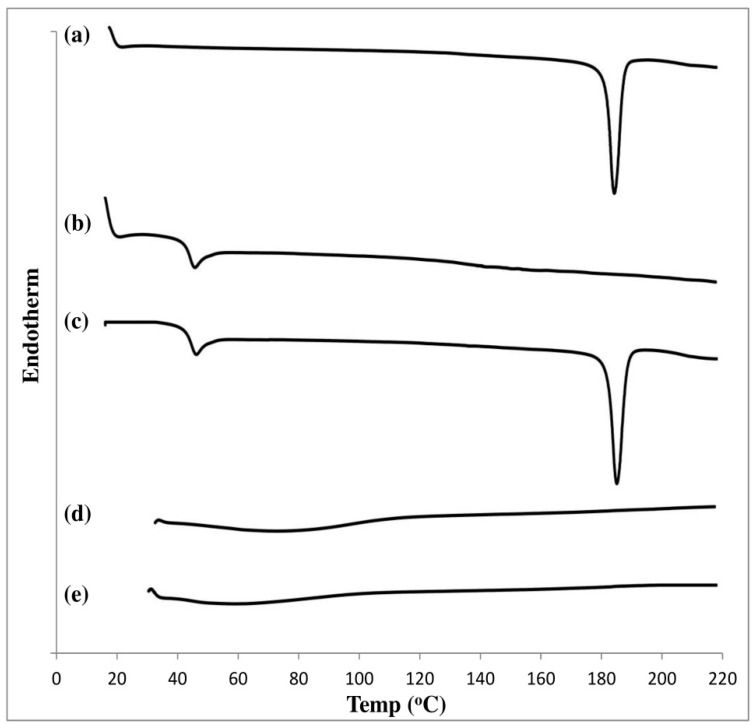
DSC thermograms of (**a**) pure alogliptin (**b**) PLGA (**c**) physical mixture (**d**) blank ISGI formulation (**e**) optimized ISGI formulation.

**Figure 11 pharmaceutics-14-01867-f011:**
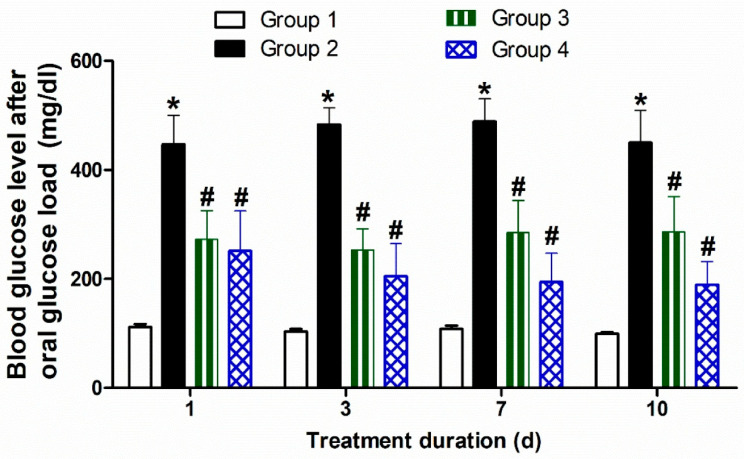
Influence of oral alogliptin solution and subcutaneous optimized ISGI formulation on blood glucose level of rats. Values are expressed as mean ± standard error of mean. * *p* < 0.05 in comparison to group I. # *p* < 0.05 in comparison to group II.

**Table 1 pharmaceutics-14-01867-t001:** Selected independent factors and dependent responses for alogliptin-loaded ISGI formulations.

Independent Factors	Unit	Symbol	Type	Actual Levels (Coded)	
				Low (−1)	High (+1)
Lactide concentration in PLGA	%	A	Numeric	65	85
Type of solvent	-	B	Categoric	DMSO	NMP
PLGA amount	mg	C	Numeric	187.5	312.5
**Dependent responses**	**Unit**	**Symbol**	**Goal**		
Burst release after 6 h	%	Y_1_	Minimize		
Cumulative release after 10 days	%	Y_2_	Maximize		

PLGA, poly (lactide-co-glycolide); DMSO, dimethyl sulfoxide; NMP, N-methyl-2-pyrrolidone.

**Table 2 pharmaceutics-14-01867-t002:** Sixteen experimental runs as suggested by 2^3^ factorial design.

Formulation *	Actual Levels			Coded Levels		
	A (%)	B	C (mg)	A (%)	B	C (mg)
**F1**	65	NMP	312.5	−1	+1	+1
**F2**	85	DMSO	312.5	+1	−1	+1
**F3**	65	NMP	312.5	−1	+1	+1
**F4**	85	DMSO	187.5	+1	−1	−1
**F5**	85	NMP	312.5	+1	+1	+1
**F6**	65	DMSO	187.5	−1	−1	−1
**F7**	65	NMP	187.5	−1	+1	−1
**F8**	65	DMSO	187.5	−1	−1	−1
**F9**	65	DMSO	312.5	−1	−1	+1
**F10**	85	DMSO	312.5	+1	−1	+1
**F11**	85	NMP	187.5	+1	+1	−1
**F12**	85	DMSO	187.5	+1	−1	−1
**F13**	65	DMSO	312.5	−1	−1	+1
**F14**	65	NMP	187.5	−1	+1	−1
**F15**	85	NMP	312.5	+1	+1	+1
**F16**	85	NMP	187.5	+1	+1	−1

A, lactide concentration in PLGA; B, type of solvent; C, PLGA amount; DMSO, dimethyl sulfoxide; NMP, N-methyl-2-pyrrolidone. * All formulations contained 62.5 mg of alogliptin.

**Table 3 pharmaceutics-14-01867-t003:** Solubility measurements of alogiptin in water-miscible and non-miscible solvents.

Solvent	Viscosity * (cP)	Molecular Volume ** (mL/mole)	Solubility of Alogliptin in Solvents (mg/mL)	Ability of Solvents to Dissolve PLGA (85:15)
**Water-miscible solvents**				
**NMP**	1.65	96.25	177.67 ± 1.54	Good
**BA**	6.00	103.98	170.12 ± 0.98	Good
**DMSO**	2.00	71.03	125.45 ± 0.39	Good
**PEG 400**	90.00	336.28–371.68	140.33 ± 1.38	Poor
**PG**	58.10	73.16	42.51 ± 1.01	Poor
**Non-miscible solvents**				
**EA**	0.423	97.68	1.12 ± 0.41	Good
**TA**	17.40	188.11	0.67 ± 0.19	Poor
**BB**	10.90	189.51	0.25 ± 0.08	Poor

NMP, N-methyl-2-pyrrolidone; BA, benzyl alcohol; DMSO, dimethyl sulfoxide; PEG 400, polyethylene glycol 400; PG, propylene glycol; EA, ethyl acetate; TA, triacetin; BB, benzyl benzoate. * Viscosity values of solvents were reported in the Handbook of Pharmaceutical Excipients [61]. ** Molecular volume was calculated by dividing the molecular weight by the density of each solvent [64].

**Table 4 pharmaceutics-14-01867-t004:** Actual and predicted values of Y_1_ and Y_2_ responses for alogliptin-loaded ISGI formulations.

Formulation	Y_1_ (%)			Y_2_ (%)		
	Actual	Predicted	Residual	Actual	Predicted	Residual
**F1**	6.28	7.55	−1.27	53.14	54.07	−0.93
**F2**	5.30	7.80	−2.50	51.92	52.85	−0.93
**F3**	8.81	7.55	1.27	54.99	54.07	0.93
**F4**	52.55	53.19	−0.64	69.35	70.30	−0.95
**F5**	3.29	4.55	−1.26	43.08	44.33	−1.25
**F6**	66.15	66.65	−0.49	78.04	78.98	−0.94
**F7**	49.04	49.52	−0.47	76.32	77.36	−1.04
**F8**	67.14	66.65	0.49	79.91	78.98	0.94
**F9**	9.19	10.29	−1.10	55.99	56.72	−0.72
**F10**	10.30	7.80	2.50	53.77	52.85	0.93
**F11**	37.26	37.91	−0.65	68.17	69.18	−1.01
**F12**	53.84	53.19	0.64	71.24	70.30	0.95
**F13**	11.39	10.29	1.10	57.44	56.72	0.72
**F14**	49.99	49.52	0.47	78.39	77.36	1.04
**F15**	5.82	4.55	1.27	45.58	44.33	1.25
**F16**	38.56	37.91	0.65	70.19	69.18	1.01

Y_1_, burst release after 6 h; Y_2_, cumulative release after 10 days.

**Table 5 pharmaceutics-14-01867-t005:** Results of ANOVA statistical analysis of Y_1_ response.

Source	Sum of Squares	df	Mean Square	F-Value	*p*-Value	Significance	% Contribution
**Model**	8712.93	7	1244.70	415.84	<0.0001	Significant	-
**A**	233.33	1	233.33	77.95	<0.0001	Significant	2.67
**B**	368.92	1	368.92	123.25	<0.0001	Significant	4.22
**C**	7838.50	1	7838.50	2618.71	<0.0001	Significant	89.72
**AB**	0.4465	1	0.4465	0.1492	0.7094	Non-significant	0.005
**AC**	95.84	1	95.84	32.02	0.0005	Significant	1.09
**BC**	174.50	1	174.50	58.30	<0.0001	Significant	1.99
**ABC**	1.39	1	1.39	0.4638	0.5151	Non-significant	0.016
**Pure error**	23.95	8	2.99	-	-	-	0.27
**Cor total**	8736.87	15	-	-	-	-	-
**Fit statistics**							
**SD**		1.73			**R^2^**		0.9973
**Mean**		29.68			**Adjusted R^2^**		0.9949
**C.V.%**		5.83			**Predicted R^2^**		0.9890
**PRESS**		95.78			**Adequate precision**		50.7598

ANOVA, analysis of variance; Y_1_, burst release after 6 h; df, degree of freedom; A, lactide concentration in PLGA; B, type of solvent; C, PLGA amount; SD, standard deviation; C.V.%, coefficient of variation; PRESS, predicted residual sum of square, R^2^, multiple correlation coefficient.

**Table 6 pharmaceutics-14-01867-t006:** Results of ANOVA statistical analysis of Y_2_ response.

Source	Sum of Squares	df	Mean Square	F-Value	*p*-Value	Significance	% Contribution
**Model**	2247.48	7	321.07	167.04	<0.0001	Significant	-
**A**	232.12	1	232.12	120.76	<0.0001	Significant	10.26
**B**	48.33	1	48.33	25.14	0.0010	Significant	2.14
**C**	1929.30	1	1929.30	1003.74	<0.0001	Significant	85.26
**AB**	7.19	1	7.19	3.74	0.0892	Non-significant	0.32
**AC**	2.64	1	2.64	1.37	0.2751	Non-significant	0.12
**BC**	17.77	1	17.77	9.24	0.0161	Significant	0.79
**ABC**	10.14	1	10.14	5.28	0.0507	Non-significant	0.45
**Pure error**	15.38	8	1.92	-	-	-	0.68
**Cor total**	2262.86	15	-	-	-	-	-
**Fit statistics**							
**SD**		1.39			**R^2^**		0.9932
**Mean**		62.97			**Adjusted R^2^**		0.9873
**C.V.%**		2.20			**Predicted R^2^**		0.9728
**PRESS**		61.51			**Adequate precision**		35.3427

ANOVA, analysis of variance; Y_2_, cumulative release after 10 days; df, degree of freedom; A, lactide concentration in PLGA; B, type of solvent; C, PLGA amount; SD, standard deviation; C.V.%, coefficient of variation; PRESS, predicted residual sum of square, R^2^, multiple correlation coefficient.

**Table 7 pharmaceutics-14-01867-t007:** Kinetic release study of optimized ISGI formulation.

Formulation	Zero Order Model	First Order Model	Higuchi Model	Hixson–Crowell Model	Korsmeyer–Peppas Model	
	R^2^	R^2^	R^2^	R^2^	R^2^	n
**Optimized ISGI** **formulation**	0.7299	0.8818	0.9807	0.8417	0.9851	0.446

R^2^, correlation coefficient; n, release exponent.

## Data Availability

Not applicable.

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
