# Peer review of "Investigation of Alogliptin-Loaded In Situ Gel Implants by 23 Factorial Design with Glycemic Assessment in Rats"

_pharmaceutics, 2022, doi:10.3390/pharmaceutics14091867_

Round 1
Reviewer 1 Report
The manuscript's authors investigated the preparation and in vivo effectiveness of an in-situ gel implant (ISGI) containing Alogliptin. During formulation, according to the design of the experiment (DoE), a three-factors, two-level full factorial design was used. Several in vitro characteristic tests (SEM, DSC, FTIR, in vitro drug release, etc.) were performed in addition to the rat investigations. The figures and tables are informative, but some corrections are necessary before publication.
- During the in vitro drug release studies (Fig. 1; 4; 9), in the case of dissolution profiles, the X-axis is not appropriate, since the first three sampling points (0.5h; 2h; 6h) represent hours, yet they are listed with the same number of units than a day, and this distorts the shape of the dissolution profile. This is not appropriate.
- The 2.2.1.2. chapter, it is mentioned that PLGA (85% lactide:15% glycolide) polymer was used with different solvents. At the same time, one of the experimental design factors is the lactide: glycolide ratio.
- In my opinion, the description of the experimental design (currently 2.2.2.) should precede the production itself (2.2.1.2.), since this will specify the composition and number of the formulas. Furthermore, in the case of a gel, viscosity is an essential property that the authors should investigate.
- The value of the applied accelerating voltage is missing from the SEM description.
Reviewer 2 Report
The manuscript pharmaceutics-1848816 presented provides the design and application of an injectable long acting PLGA based in situ gel implant loaded with alogliptin for the treatment of diabetes type 2.
Although the technique and experimental approach are not original, the manuscript has the merit to detail the experimental approach leading to the formulation that was finally used for the in vivo application. The English is reasonable, but there are some sections that need to be improved for better understanding. I suggest that the whole manuscript is revised by a native English speaker. Overall, I consider that the manuscript is not well balanced because it is overdetailed in the sections regarding the factorial design and analysis of the different factors that affect the formulation performance to a level that leaves the readers confused, and in the end, only one small section (3.4.4) is dedicated to the in vivo application. A total of 13 figures for a research article also seems excessive to me. The authors should consider the transfer of some of this information to supplementary data and detail more the in vivo results.
Specific Comments
- Introduction (page 2) – the authors refer that the production of nanoparticles based formulations are expensive, complex and lack stability. This is controversial, given that there are several reports in the literature suggesting the opposite. In fact, one of the merits of the nanoparticles technology is the ease of manufacturing and scale up production, as well as the stability of the formulations.
- Section 2.2.1.1 and 2.2.1.3– the authors used a spectrophotometer to quantify alogliptin. Given that this method is very susceptible to interferences, did the authors perform the necessary controls?
- section 2.2.5.1 – the unit for weight is g and not gm.
- figures 2, 3 and 6 – It seems that the figures are a simple copy from the software. They are very confusing, with some text that is over the bars and can’t be read. These figures need to be revised and simplified.
- figure 7 needs to be revised. The letter size is too small and legends are very hard to read.
Round 2
Reviewer 1 Report
Thanks you for your answers. I accept those and the corrections made in the manuscript.
